# Enhanced Membrane Incorporation of H289Y Mutant GluK1 Receptors from the Audiogenic Seizure-Prone GASH/Sal Model: Functional and Morphological Impacts on *Xenopus* Oocytes

**DOI:** 10.3390/ijms242316852

**Published:** 2023-11-28

**Authors:** Sandra M. Díaz-Rodríguez, Isabel Ivorra, Javier Espinosa, Celia Vegar, M. Javier Herrero-Turrión, Dolores E. López, Ricardo Gómez-Nieto, Armando Alberola-Die

**Affiliations:** 1Neuroscience Institute of Castilla y León (INCyL), University of Salamanca, E-37007 Salamanca, Spain; sdiazrodriguez@usal.es (S.M.D.-R.); mjaviht@usal.es (M.J.H.-T.); richard@usal.es (R.G.-N.); 2Institute of Biomedical Research of Salamanca (IBSAL), E-37007 Salamanca, Spain; 3Department of Physiology, Genetics and Microbiology, University of Alicante, E-03690 Alicante, Spain; isabel.ivorra@ua.es (I.I.); javier.espinosa@ua.es (J.E.); celia.vegar@ua.es (C.V.); alberoladie.armando@ua.es (A.A.-D.); 4Neurological Tissue Bank INCYL (BTN-INCYL), University of Salamanca, E-37007 Salamanca, Spain

**Keywords:** epilepsy, GluK1, genetic variant, kainate currents, membrane incorporation, *Xenopus* oocytes

## Abstract

Epilepsy is a neurological disorder characterized by abnormal neuronal excitability, with glutamate playing a key role as the predominant excitatory neurotransmitter involved in seizures. Animal models of epilepsy are crucial in advancing epilepsy research by faithfully replicating the diverse symptoms of this disorder. In particular, the GASH/Sal (genetically audiogenic seizure-prone hamster from Salamanca) model exhibits seizures resembling human generalized tonic-clonic convulsions. A single nucleotide polymorphism (SNP; C9586732T, p.His289Tyr) in the *Grik1* gene (which encodes the kainate receptor GluK1) has been previously identified in this strain. The H289Y mutation affects the amino-terminal domain of GluK1, which is related to the subunit assembly and trafficking. We used confocal microscopy in *Xenopus* oocytes to investigate how the H289Y mutation, compared to the wild type (WT), affects the expression and cell-surface trafficking of GluK1 receptors. Additionally, we employed the two-electrode voltage-clamp technique to examine the functional effects of the H289Y mutation. Our results indicate that this mutation increases the expression and incorporation of GluK1 receptors into an oocyte’s membrane, enhancing kainate-evoked currents, without affecting their functional properties. Although further research is needed to fully understand the molecular mechanisms responsible for this epilepsy, the H289Y mutation in GluK1 may be part of the molecular basis underlying the seizure-prone circuitry in the GASH/Sal model.

## 1. Introduction

Ionotropic glutamate receptors are ligand-gated ion channels permeable to cations which mediate most of the excitatory synaptic transmission in the central nervous system. They are divided into four functional classes, attending to their sequence identity and pharmacological properties: (i) α-amino-3-hydroxy-5-methyl-4-isoxazolepropionic acid (AMPA) receptors, (ii) kainate receptors (KARs), (iii) N-methyl-d-aspartate (NMDA) receptors, and (iv) glutamate delta (GluD) receptors [1,2,3,4]. More specifically, KARs are predominantly arranged in homomeric or heteromeric combinations of five different pore-forming subunits, known as GluK1-5, encoded by the KAR type 1-5 genes (*Grik1*-*5*) [5,6]. The low-affinity (named for the relative low affinity and agonist potency for glutamate) GluK1-3 subunits can assemble as functional homotetramers and heterotetramers, while the high-affinity GluK4/5 subunits can only form functional heterotetrameric receptors with GluK1-3 subunits [3,6]. All KAR subunits have a shared architecture composed of (i) a large extracellular amino-terminal domain (ATD), which participates in assembly, trafficking, and the functional regulation of receptors; (ii) a ligand-binding domain (LBD); (iii) a transmembrane domain (TMD), comprising four membrane spanning-segments which form the cation-selective ion channel; and (iv) a cytoplasmic C-terminal domain (CTD), which mediates receptor localization and intracellular regulation [1,7,8,9,10,11]. Alternative splicing of GluK1 CTD can produce four different isoforms named GluK1-a, GluK1-b, GluK1-c [12], and GluK1-d (which is only found in humans [13,14]). GluK1-b is a larger isoform than GluK1-a, comprising an endoplasmic reticulum (ER) retention motif, whereas GluK1-c includes an additional sequence of 29 residues into the CTD of GluK1-b [15]. ATD can also undergo alternative splicing, resulting in GluK1-1 or GluK1-2 isoforms, the former with a 15-residue insertion in this domain. Moreover, RNA editing at the Q/R site, located in the M2 pore-loop of the TMD, drastically changes the properties of the ion channel. Thus, the unedited receptor GluK1(Q) is sensitive to polyamine channel-block and permeable to Ca^2+^, unlike the edited form GluK1(R) [12,16].

KARs can localize in presynaptic neurons, modulating the release of excitatory and inhibitory neurotransmitters; in postsynaptic neurons, promoting excitatory neurotransmission; and in extrasynaptic ones, contributing to neuronal development and plasticity [2,17,18]. Remarkably, since these receptors have a pivotal role in excitatory neurotransmission, they constitute a key therapeutic target, as their dysfunction causes relevant disorders. Indeed, several alterations in *Grik* genes have been related to Alzheimer’s disease, Huntington’s chorea, amyotrophic lateral sclerosis, and epilepsy (reviewed in [1,19]). The latter is a chronic neurological disease with a multifactorial origin, including genetic mutations, and is defined by the appearance of sudden and recurrent seizures. Epilepsy has high epidemiological implications worldwide and, therefore, it is important to expand the knowledge on the genetic and functional basis of epilepsy to increasingly discover the inheritance patterns, genetic heterogeneity, and functional mechanisms responsible for this pathology [20]. Animal models of epilepsy serve as indispensable tools for discovering genes and pathways associated with seizures, offering potential avenues for the development of targeted treatments for this condition [21]. A noteworthy example is the genetically audiogenic seizure-prone hamster from Salamanca (GASH/Sal) strain, which exhibits sound-induced seizures (allowing a precise control of them by the experimenter) similar to the generalized tonic–clonic seizures observed in epileptic patients [22,23,24]. Recently, a whole-exome sequencing in GASH/Sal animals has been performed to identify and characterize the mutational landscape of this strain [25]. It has been described, among other findings, that GASH/Sal hamsters have a moderate-impact variant in the *Grik1* gene that causes a single-point mutation consisting of the substitution of a histidine for a tyrosine at residue 289 (SNP; C9586732T, p.H289Y), located in the ATD of GluK1 receptors (GluK1Rs) [25]. The important role played by GluK1Rs at excitatory synapses in the central nervous system might suggest their involvement in epileptogenesis [26]. Indeed, *Grik1* has been identified previously as an epilepsy-associated gene [27,28], and it has been described in relation to *Grik1* variants with juvenile absence epilepsy [29] and with sudden unexpected death in epilepsy [30]. However, to our knowledge, no epilepsy-related mutations located in the ATD of GluK1Rs have been reported.

In the present work, we studied, using confocal immunofluorescence microscopy, the effects of the single-point mutation H289Y in comparison with their wild-type (WT) on the expression and incorporation to the membrane of GluK1-2aRs that were heterologously expressed in *Xenopus laevis* oocytes. Moreover, to decipher the functional effects of this mutation, we used the two-electrode voltage-clamp technique, which is a well-established, highly robust, and versatile method for the functional evaluation of membrane proteins expressed in oocytes [31].

## 2. Results

### 2.1. Assessing the Impact of the H289Y Mutation on the Function and Stability of GluK1Rs using Sequence-Homology Based Tools

The H289Y mutation in GluK1R was subjected to analysis using the sequence homology-based tools Sorting Intolerant From Tolerant (SIFT) and iStable. Both are computational predictors that allowed us to evaluate the impact of the H289Y mutation on the function and thermodynamic stability of GluK1Rs. As a result, the prediction analysis for the H289Y mutation suggested that it was tolerant (SIF score > 0.05) and did not decrease the stability of the protein (iStable conf. score = 0.56). 

The results obtained with these kinds of tools are a prediction and constitute a complementary instrument that cannot substitute experimental data obtained in the laboratory. Indeed, SIFT has been reported to perform well in predicting loss-of-function variants [32]. However, the ability to accurately identify variants with a normal function or gain-of-function is inferior, so that more than 40% of variants with altered receptor function demonstrated using biological tests were falsely predicted as benign [32]. Therefore, the expression, localization, and function of WT and H289Y GluK1Rs was evaluated using confocal immunofluorescence microscopy and electrophysiological techniques in *Xenopus* oocytes.

### 2.2. Expression and Localization of GluK1 Receptors in Xenopus Oocytes

Initially, we microinjected into *Xenopus* oocytes two expression vectors (WT, *Grik1*_WT; and with the mutation, *Grik1*_H289Y; see Figure 1) that contained the complete *Grik1* mRNA sequence, and we did not detect any type of immunolabeling of the corresponding translated GluK1 proteins (WT_GluK1-2b and H289Y_GluK1-2b; Figure 1). The work published by Han et al. [15] allowed us to find out that these GluK1s amino acid sequences translated with the microinjected *Grik1*-mRNAs contained a retention region in the ER and, consequently, GluK1-2bRs could not reach the oocyte plasma membrane. Based on these data, we microinjected two other types of expression vectors for these types of GluK1 proteins, but this time without containing in its sequence the ER retention region (*Grik1*_WT_nER and *Grik1*_H289Y_nER; Figure 1). On these experimental groups, we detected immunolabeling of these types of GluK1 proteins. Thus, both WT GluK1-2aR and H289Y GluK1-2aR reached the oocyte plasma membrane.

Then, we assessed the localization and distribution of WT and H289Y GluK1-2aRs throughout sections of whole oocytes, including the analysis of the distribution of receptors in the animal or vegetal hemispheres of the cells. In these sections, a lower immunoreactivity of WT GluK1-2aRs compared to H289Y GluK1-2aRs was detected (Figure 2A,B). Furthermore, the analysis of the immunoreactivity of GluK1-2aRs using RawIntDen in MATLAB matched the architecture of *Xenopus* oocytes. Therefore, our MATLAB mapping procedure provided a suitable reference for a visual analysis of the GluK1-2aRs’ distribution (Figure 2C,D). Regarding the digital mapping of the oocyte-membrane expressing H289Y GluK1-2aRs, an increase in the red scale (larger staining) was calculated in the area of both the animal and vegetal hemispheres in comparison to the expression of WT GluK1-2aRs. The RawIntDen dataset is visually presented in Figure 2E to illustrate the quantitative distinctions between WT GluK1-2aRs and H289Y GluK1-2aRs, demonstrating a statistically significant difference (5654.0 ± 19.5 mean number of particles for WT GluK1-2a vs. 28764.0 ± 173.6 mean number of particles for H289Y GluK1-2a; *p* < 0.01). Therefore, there was an accumulation of labeling near the surface of the oocyte in both the animal and vegetal poles, suggesting a selective accumulation of H289Y GluK1-2aRs in this area, showing significant differences between the WT GluK1-2aRs compared to the H289Y GluK1-2aRs.

Then, using a confocal laser microscope, we conducted a rigorous quantification of GluK1-2aR-immunoreactivity (for both the WT and the mutant) within the *Xenopus* oocyte membrane and the ER proximal to the oocyte membrane. Notably, we observed a significant decrease in immunoreactivity for WT GluK1-2aR (Figure 3A–F) when compared to the H289Y GluK1-2aR (Figure 3G–K) in both the animal (Figure 3M; 2845.0 ± 267.7 a.u. for WT GluK1-2a vs. 6030.0 ± 711.6 a.u. for H289Y GluK1-2a; *p* < 0.01) and vegetal hemisphere membranes (Figure 3N; 2543.0 ± 276.4 a.u. for WT GluK1-2a vs. 4879.0 ± 281.5 a.u. for H289Y GluK1-2a; *p* < 0.01). Furthermore, it is noteworthy that the immunoreactivity of H289Y GluK1-2aR seemed to exhibit a distinct pattern, showing a continuous overlay on the membrane, in contrast to the WT GluK1-2Ra, which displayed well-defined spherical accumulations of approximately 0.3 µm (Figure 3C,F,I,L). Additionally, we identified discernible differences between the animal and vegetal hemispheres within each experimental group. Remarkably, in both experimental groups (WT and mutated receptors), the expression of GluK1-2aRs was higher in some points of the animal hemisphere when shown juxtaposed with the vegetal hemisphere (Figure 3O,P; values at 140 μm for WT and H289Y, and values at 80, 120, and 140 μm for H289Y).

### 2.3. The H289Y Mutation Enhances Kainate Currents in Oocytes Expressing GluK1-2aRs

The membrane conductance of uninjected oocytes was unaffected by bathing the cells for 5 s with Ka (up to 500 μM) while holding the membrane potential at −60 mV, in accordance with previous data [33]. Similarly, no Ka currents (*I_Ka_*s) were evoked in oocytes previously microinjected with mRNAs coding for WT and H289Y GluK1-2b as expected, due to the presence of the ER retention motif in these mRNAs [15]. However, the application of 100 µM Ka for 5 s elicited an inward *I_Ka_* (Figure 4A, WT_GluK1-2a, black recording) in oocytes previously microinjected with WT GluK1-2a-encoding mRNA. Thus, these WT GluK1-2aRs were functionally incorporated in the membrane of 23 oocytes out of 51 (from four different donors), that is, 45.1% of the cells. A statistically significant increase in functional expression (*χ*^2^-test, *p* < 0.05) was observed when the oocytes were microinjected with H289Y GluK1-2a-encoding mRNA. Actually, *I_Ka_*s were evoked in 30 cells of 46 (63.2%; from four donors); see a representative *I_Ka_* in Figure 4A (H289Y_GluK1-2a, red recording). The average amplitudes of the *I_Ka_*s from oocytes incorporating either WT or H289Y GluK1-2a receptors were of −37.8 ± 8.3 nA and −61.0 ± 11.0 nA, respectively. No significant differences were found between both groups (Mann–Whitney rank-sum test, *p* > 0.05), most likely due to the large variability in *I_Ka_*s among oocytes from different donors. For this reason, the *I_Ka_* amplitude for each oocyte was normalized to the maximum *I_Ka_* evoked in its corresponding donor. As shown in Figure 4B, the average percentage of the maximum *I_Ka_* was significantly larger for oocytes that incorporated H289Y GluK1-2a receptors (40.5 ± 5.9%, red column; Mann–Whitney rank-sum test, *p* < 0.05) than those expressing WT GluK1-2a receptors (22.7 ± 4.1%, black column).

### 2.4. The H289Y Mutation Does Not Significantly Modify the Functional Properties of GluK1-2aRs

Firstly, to assess the possible effects of the H289Y single-point mutation in the functional properties of GluK1-2aRs, *I_Ka_*s were obtained through the subsequent superfusion of different concentrations of Ka (1, 5, 10, 50, 100, and 500 µM) to oocytes bearing either WT (Figure 5A, black recordings) or H289Y (Figure 5A, red recordings) GluK1-2aRs. Figure 5B shows the relationship between the Ka concentration and *I_Ka_* amplitude for WT (black solid symbols) and H289Y (red solid symbols) GluK1-2aRs after fitting the Hill equation curve to the experimental data. The estimated *EC_50_* and *n_H_* values of the fitted curve were 23 µM (CI = 9–26 µM) and 1.3 ± 0.3 for WT GluK1-2aR, respectively, and 12 µM (CI = 10–16 µM) and 1.3 ± 0.2 for H289Y GluK1-2aRs, respectively. Therefore, despite the slight shift to the left of the curve of H289Y Gluk1-2aR, which could indicate an increase in the affinity and/or efficacy of Ka for these receptors, this was not a statistically significant effect.

Furthermore, we obtained current–voltage (I–V) relationships at the steady-state component of *I_Ka_*s. For this, voltage jumps (from −120 to +60 mV, in 20 mV steps) were imposed on oocytes bearing either WT (Figure 6A, black recordings) or H289Y (Figure 6A, red recordings) GluK1-2aRs, while superfusing normal Ringer (NR) solution at the *I_Ka_* plateau elicited by adding 100 µM Ka to the NR solution. The I–V curves of net *I_Ka_*s elicited in both WT (Figure 6B, solid black symbols) and H289Y (Figure 6B, solid red symbols) GluK1-2aR groups showed a reversal potential close to 0 mV, indicating that channel-permeability properties were unaffected by the single point mutation H289Y. Likewise, the characteristic inwardly rectifying I–V shape of unedited forms of GluK1-2aR was maintained.

## 3. Discussion

An SNP in the *Grik1* gene (C9586732T, p.H289Y) [25] has been previously reported in GASH/Sal hamsters, causing the GluK1Rs encoded by that gene to show the H289Y mutation located in the ATD (shown in Figure 1). Thus, in the present work, we evaluated the effects of this mutation on GluK1-2aRs through their heterologous expression in *Xenopus* oocytes. These receptors, after the microinjection of their coding mRNA, were expressed by the oocytes and incorporated in their membrane as homotetramers. We found differences between WT and H289Y GluK1-2aRs in terms of their incorporation and targeting to the oocyte membrane, as well as in the amplitude of the responses evoked with Ka, their agonist. The main results were as follows: (i) the H289Y mutation increases the expression and incorporation of GluK1-2aRs into an oocyte’s membrane, as determined using confocal immunofluorescence microscopy, and this was higher in the animal hemisphere; (ii) this outcome was consistent with the increase in the functional expression and the percentage of maximum *I_Ka_* obtained using the two-electrode voltage-clamp technique in oocytes bearing H289Y GluK1-2aRs; (iii) to our knowledge, we have characterized, for the first time, some of the electrophysiological properties of the GluK1-2aR from *Mesocricetus auratus*; and (iv), as predicted using sequence-homology based tools, the H289Y mutation does not modify the functional properties of GluK1-2aRs, and this result was confirmed in accordance with the results obtained from dose–response curves and I–V relationships.

Most of the disease-associated genetic variants affecting human *Grik* genes have been described for the *Grik2* gene, altering ATD (five variants) and TMD-linker domains (seven variants), while only two variants have been described for the *Grik1* gene, both of which affect the CTD [1]. Our results suggest that the H289Y gain-of-function mutation, located in the ATD of the GluK1-2aR from the GASH/Sal hamster, may be related with an important mechanism for the control of GluK1-2aR surface expression and, in particular, for regulating GluK1-2aR levels at synapses, even though this should be demonstrated in further research. Indeed, although many previous studies have already indicated the importance of the intracellular CTD for KARs trafficking and retention in the ER [34,35,36,37], as stated above (see Section 1), it is known that alterations in the ATD would contribute to changes in the regulation of the synaptic and surface expression of KARs [7]. However, the scarce studies describing the role of ATD regulating the membrane trafficking of KARs have been carried out by replacing a large number of amino acids in the ATD or even the whole domain [7,10], but not by a single-point mutation. Studies carried out in CA1 neurons hippocampus showed that the GluK1 receptors are completely excluded from synapses when a signal peptide interacts with ATD, but the replacement of either a signal peptide or the ATD domain with the corresponding GluK2 sequences enables GluK1 to appear on the neuronal surface and at the synapse [7]. Further research is needed to understand in detail the molecular mechanisms through which the H289Y mutation increases the expression and incorporation of GluK1-2a receptors into the plasma membrane. This should include studies to analyze the functional relevance of this mutation in a native system to know whether the increased protein expression will contribute to the diseased phenotype or not. Other studies are needed to decipher the possible effects of the H289Y mutation in the GluK1-2a receptor and its interaction with certain proteins, such as Neuropilin Tolloid-like 1 and Neuropilin Tolloid-like 2 (NETO1 and NETO2, respectively), among others, since GluK1 and GluK2 ATDs have been described to have a differential dependence on NETO proteins, which regulate extrasynaptic and synaptic trafficking [2,10].

In relation with the distribution of GluK1-2aRs in an oocyte’s membrane, we observed that there was a higher presence of receptors in some points of the animal hemisphere plasma membrane. Oocytes were microinjected with mRNAs coding for GluK1-2aRs in the vegetal hemisphere close to the equator, as this is the best procedure reported [38]. It is known that native muscarinic acetylcholine receptors, ρ1 γ-aminobutyric acid (GABA) receptors, α7 nicotinic receptors, and GluA3 glutamate receptors expressed in the oocyte plasma membrane after the injection of their respective mRNAs, are all located mainly in the animal hemisphere [39,40,41]. However, cat and *Torpedo* nicotinic receptors, incorporated in oocytes after mRNA microinjection, were preferentially inserted in the vegetal site [42]. Besides these heterogeneous expressions in the oocyte plasma membrane, which were more often in the animal hemisphere, another reported difference in the location has been whether the heterologous proteins were evenly distributed or by contrast were found in patches. In fact, when injecting membranes containing nicotinic receptors, they were incorporated in the oocyte membrane in patches; in contrast, a homogeneous distribution was found when the same receptors were incorporated after the injection of the corresponding mRNA [31,38]. The differential distribution of receptors between oocyte’s hemispheres, like those now reported for WT and H289Y GluK1-2aRs, could be due to an uneven distribution of the translation machinery and subsequent translocation of the proteins to the plasma membrane, or to a pre-existence of some components which facilitate the binding of receptors to the plasma membrane [42].

As stated above, the H289Y mutation does not modify the functional properties of WT GluK1-2aRs, but this fact is not unexpected as the ATDs in KARs seem to minimally regulate their function, mainly due to the limited ATD–LBD interaction [1,43]. However, an improvement in the functional expression of mutated GluK1-2aRs in oocytes was observed, which was shown as an increase in the percentage of maximal *I_Ka_*; this was in accordance with the results obtained using confocal immunofluorescence microscopy. Furthermore, the electrophysiological features of both WT and H289Y GluK1-2aRs from *M. auratus* resembled those of GluK1-2aRs from other species: (i) the *EC*_50_ values obtained in the dose-response curves matched those previously obtained by other authors. Sommer et al. [12] and Alt et al. [44] reported an *EC*_50_ of 33.6 and 21.3 µM using a whole-cell patch-clamp technique in HEK293 cells that transiently expressed rat and human GluK1-2aRs, respectively. (ii) We obtained a Hill coefficient (*n_H_*) close to 1 for hamster GluK1-2aRs, suggesting that the binding of a single molecule of Ka to the LBD of a GluK1-2aR is enough to open its ion channel, in strong concordance with previously published data [44] (*n_H_* = 0.82 ± 0.09). (iii) The hamster GluK1-2aRs’ I–V relationships showed the characteristic inwardly rectifying shape of unedited forms of GluK1-2aRs, which are sensitive to a polyamine blockade due to the presence of a glutamine (Q) within the pore, unlike when it is absent and a charged arginine is present in edited forms of GluK1-2aRs [12,16]. (iv) The reversal potential at 0 mV indicated that the ion channel was also similar to that described previously [12].

Several genetic variants associated with disease in humans have been described for *Grik* genes such as schizophrenia, intellectual disability, bipolar and movement disorders, autism, and epilepsy [1]. It is established that epilepsy is caused by an imbalance between inhibitory and excitatory neuronal signaling within the brain [45]. The main neurotransmitter mediating the excitatory conductance of nerve cells in the central nervous system is glutamate, which binds to AMPARs and NMDARs, which are present at most glutamatergic synapses, or KARs, which are much more selectively expressed, e.g., GluK1Rs are mainly expressed in the hippocampus, cortical interneurons, Purkinje cells, and sensory neurons [7,19,46]. Moreover, it has been described that KARs are related to epileptogenesis, as the net effect of their activation in vivo causes a decrease in GABA release, which, in turn, leads to an increase in excitability and thus epileptiform activity [47]. In this context, the use of the GASH/Sal strain as an animal model to study epilepsy is fundamental because it enables progress in the knowledge of the cellular and molecular bases that cause this disease in the animal under study, but also because it could be used to determine mechanisms equivalent to those that occur in humans [24].

In addition to the study presented here, our research group conducted in silico experiments utilizing protein 3D-structure modelling and various computational stability predictors. These experiments indicated, in accordance with the in silico predictions and functional data obtained in the present work, that the single-point mutation H289Y likely leads to protein stabilization by increasing the number of intermolecular interactions, as compared to the wild-type GluK1 receptor [48]. Concurrently, we observed a concentration of GluK1 immunoreactivity near the cell nucleus in brain structures associated with the GASH/Sal seizure neuronal network [48]. These combined findings support the assertion made in our study that the mutation has the potential to alter the GluK1 trafficking mechanism while exerting a relatively minor influence on the functional properties of the receptor. The precise genetic alterations and molecular mechanisms contributing to audiogenic susceptibility in the GASH/Sal strain remain an ongoing area of investigation, necessitating further research. Nevertheless, our present study unequivocally demonstrates an increase in the expression and incorporation of functional GluK1 receptors into the oocyte’s membrane, and this finding is supported by our immunolabeling and electrophysiological data. Probing the existence of functional receptors incorporated into the membrane is of paramount importance in determining whether the mutation present in the GASH/Sal hamster could have a genuine implication for epileptogenesis and other techniques, such as immunoblotting, which is used as a valuable tool for confirming protein levels, cannot surpass that given by the two techniques we have used in this work. On that basis, our hypothesis centers on the notion that the H289Y mutation in GluK1-2aRs may lead to an increased trafficking of these receptors to neuronal membrane sites within the circuit responsible for seizures, thereby favouring an excitatory imbalance of the neuronal activity in critical brain regions within the GASH/Sal model.

## 4. Materials and Methods

### 4.1. In Silico Prediction Tools

Two highly reliable in silico prediction tools, SIFT version 6.2.1 and iStable, were employed to evaluate the potential functional impact of the H289Y mutation on the GluK1 receptor. SIFT 6.2.1 assesses evolutionary conservation, predicting the tolerability or adverse effects of amino acid substitutions, thereby shedding light on functional alterations. Additionally, iStable utilizes machine-learning algorithms to forecast stability changes resulting from mutations, providing invaluable insights into structural modifications. These tools were specifically selected for their robustness in predicting both functional and stability alterations induced by mutations. The in silico assessment utilized the online sequence homology-based tools: SIFT version 6.2.1 (accessible at https://sift.bii.a-star.edu.sg/index.html, accessed on 8 November 2023) and iStable (accessible at http://predictor.nchu.edu.tw/iStable/, accessed on 8 November 2023).

### 4.2. Design of the Vector and Synthesis of mRNA Coding for Wild-Type and Mutant GluK1 Receptors

A cDNA fragment of *M. auratus* encompassing the gene encoding the GluK1 receptor (GenBank accession XM_005073843.3, *Grik1* subunit 1 isoform X1) was synthesized and cloned using GenScript Biotech (Rijswijk, The Netherlands). *Grik1* cDNA fragments, bearing either the WT sequence or the missense point mutation H289Y, were both synthesized and cloned into the pcDNA3.1 vector (Invitrogen, Waltham, MA, USA) using GenScript gene synthesis service, and yielded the plasmids pcDNA_*Grik1*_*WT* and pcDNA_*Grik1*_*H289Y*, respectively. The final constructs were verified using automated dideoxy DNA sequencing.

The plasmids were used as templates to obtain, using primer pairs T7ultra + Glu2R or T7ultra + Glu-trunca-1R, four PCR products: full-length *Grik1*_*WT* and *Grik1*_*H289Y*, and two truncated derivatives lacking the region coding for the ER retention motif described previously, namely, *Grik1*_*WT_nER* and *Grik1*_*H289Y_nER* [15,49,50,51,52].

Each of these purified PCR amplicons was then employed as a template in an in vitro transcription reaction using the mMessage mMachine T7 Transcription kit (Invitrogen, Waltham, MA, USA, AM1340) following the manufacturer’s recommendations. After visualization through agarose gel electrophoresis, transcripts were subjected to polyadenylation using the Poly(A) Tailing kit (Invitrogen, Waltham, MA, USA, AM1350) to enhance stability. The effective increase in size for all four mRNA transcripts was verified through gel electrophoresis. Finally, a purification step was carried out using the MEGAclear™ Transcription Clean-Up Kit (Invitrogen, Waltham, MA, USA, AM1909) and quantification of purified mRNA was performed using Qubit (Invitrogen, Waltham, MA, USA).

### 4.3. Oocyte Microinjection with mRNA Coding for GluK1 Receptors

Adult female *Xenopus laevis* (purchased from European *Xenopus* Resource Centre at the University of Portsmouth, Portsmouth, UK) were immersed, during 15–20 min, in 0.17% tricaine methanesulfonate (MS-222) and a piece of ovary was removed under aseptic conditions. The study was conducted in accordance with the guidelines for the care and use of experimental animals adopted by the European Union (European Communities Council Directive of 22 September 2010, 2010/63/UE), and the animal protocol was approved by the Ethic Committee of Universidad de Alicante (protocol codes 2019/VSC/PEA/0097 type 2 from 25 April 2019, and 2023-VSC-PEA-0114 type 2 from 31 May 2023). The surrounding layers of isolated Dummont stage V and VI oocytes were removed manually with tweezers. Before use, cells were maintained inside an incubator at 15–16 °C in a modified Barth’s solution (88 mM NaCl, 1 mM KCl, 2.40 mM NaHCO_3_, 0.33 mM Ca(NO_3_)_2_, 0.41 mM CaCl_2_, 0.82 mM MgSO_4_, 10 mM HEPES (pH 7.4), 100 U/mL penicillin, and 0.1 mg/mL streptomycin). A total of 50 nL of solution, containing 30 ng of WT or H289Y GluK1-2a or GluK1-2b mRNAs, was microinjected in oocytes.

### 4.4. Oocyte Preparation, Processing, Immunostaining, and Imaging

#### 4.4.1. Immunohistochemistry

Firstly, oocytes previously microinjected with mRNAs coding for full-length and truncated WT and mutant GluK1 receptors were fixed in 0.1 M phosphate-buffered saline (PBS) with 4% paraformaldehyde. Cells were then stored in this PBS until being embedded in paraffin. The oocytes were then serially sectioned at 10 µm using a sliding microtome (the total number of slides per cell was 12). Slides containing oocyte sections in the equatorial plane were selected for histological processing.

After selection of the slides, deparaffinization and antigen recovery were performed at 90 °C in 10 mM sodium citrate buffer at 0.05% with Tween 20 (pH 6.0). Slides were washed in Tris-buffered saline solution (TBS 0.5 M and then TBS 0.05 M), continuing with endogenous peroxidase inactivity. The samples were blocked for 2 h with 6% normal goat serum (Catalog No. S –1000, Vector Labs) in TBS-Tx dilutions of antisera in TBS 0.05 M containing Triton X-100 at 0.1% (Catalog No. T9284; Sigma, St. Louis, MO, USA). The washes were again realized with the TBS 0.5 M and then TBS 0.05 M. This technique included a simple procedure to quench autofluorescence of *Xenopus* and to remove surface pigment from oocytes which may have interfered with fluorescence imaging [53,54]. We followed a procedure previously described [55] to perform the preparation of 1% H_2_O_2_, 5% formamide (Catalog N°199837, Merk, Darmstadt, Germany) and 0.5 × SSC (0.75 M NaCl, 0.075 M Sodium Citrate) (Catalog N°51804, Sigma; Catalog N°7647-14-5, Panreac, Barcelona, Spain). To decolorize the oocytes they were depigmented in the photobleaching solution for 2 h under fluorescent light (~488–570 nm). Subsequently, the samples were washed in TBS 0.5 M and then TBS 0.05 M, and incubated with the primary antibodies rabbit anti-GluK1 (Catalog N°ab118891; Abcam, Cambridge, UK) in TBS-T × 0.1% at 1:1000 dilution for 72 h at 4 °C. This antibody has been effectively employed in our ongoing research on the GASH/Sal brain [48]. However, despite the fact that their reactivity was not tested in oocytes, a multisequence alignment analysis of GluK1 of distinct species demonstrated the high conservation of epitopes. The sections were then washed and this was followed by incubation with Alexa Flour^TM^ 647 Goat anti-Rabbit IgG (catalog N°A21244; ThermoFisher, Waltham, MA, USA) at 1:750 dilution for 2 h. Finally, Vectashield^®^ mounting medium was used to cover the sections and preserve their fluorescence, containing DAPI stain (4,6-diamidino-2-phenylindole, catalog N°H-1200, Vector Labs., Newark, CA, USA). Negative controls were not treated with primary antibodies, and this resulted in no immunolabeling.

#### 4.4.2. Observation and Study of Histological Samples

A Leica Stellaris confocal laser coupled to a Leica DMI8 microscope (Leica Microsystems, Wetzlar, Germany) was used to study the sections processed for immunofluorescence. DyLight^®^ 647 (red) and DAPI (violet) fluorochromes (Invitrogen, Waltham, MA, USA) were detected sequentially, stack by stack, using the 647 and 405 nm laser spectral lines, respectively, and an acousto-optic beam splitter as a tunable dichroic filter system. In addition, the following lenses were used: ×4, ×10, and oil-immersion ×63/numerical aperture 1.40. With the aim of generate a maximal intensity z projection of stacks, a series of 10–15 confocal images was obtained to determine the distribution of the immunolabeled terminals.

### 4.5. Mapping of WT and H289Y GluK1-2a Proteins in Xenopus Oocytes

#### 4.5.1. Image Analysis

The image analysis was conducted using ImageJ software (version 1.51g-v1.51n; Fiji package), an open-source tool routinely employed by our research group [56]. In this current image analysis, a comprehensive assessment of various parameters was undertaken. This encompassed the quantification of the product of area and mean gray value (IntDen), the summation of pixel values (RawIntDen), determination of particle area, and computation of mean and mode values for both X and Y coordinates. To ensure uniform visual comparisons among samples, all image processing operations were executed simultaneously. Notably, the original captured images remained unaltered in terms of brightness and contrast, preserving the integrity of the data.

#### 4.5.2. MATLAB Maps

Employing MATLAB software (© MATLAB R-2017, MathWorks, Natick, MA, USA), extensive full-sections maps which encoded the location, area, and density of segmented particles were made. Using *Scatterplot* function and the MATLAB color scale “Hot”, the RawIntDen was converted to color.

### 4.6. Two-Electrode Voltage-Clamp Recordings in Xenopus Oocytes

As previously described [57], oocyte membrane currents were recorded 48–72 h after mRNA microinjection. The solution that continuously superfused oocytes was a normal frog Ringer’s (NR) solution composed of 2 mM KCl, 1.8 mM CaCl_2_, 115 mM NaCl, and 5 mM HEPES, and it had pH 7.0. Unless otherwise stated, the membrane potential of oocytes was clamped at −60 mV. The currents elicited by Ka (*I_Ka_*s) were sampled at five-fold the filter frequency (Digidata series 1440A and 1550; Axon Instruments, Foster City, CA, USA), low-pass filtered at 30–1000 Hz, and then recorded on two PCs using two appropriate software, WCP v. 4.8.6 (Strathclyde Electrophysiology Software, University of Strathclyde, Scotland, UK) and AxoScope v. 10.0.0.60 (Molecular Devices Corporation, Sunnyvale, CA, USA).

### 4.7. Experimental Design and Data Analysis of Functional Experiments

To characterize the functional profile of either WT or H289Y GluK1-2a receptors, Ka concentration–*I_Ka_* amplitude curves were obtained by bathing GluK1-2a receptor-bearing oocytes with increasing Ka concentrations. *I_Ka_*s were normalized to the maximum *I_Ka_* and a sigmoid curve was fitted to these values (see equation 1 below). An interval of at least 3 min between consecutive Ka applications was maintained in order to avoid GluK1-2aRs desensitization. The current–voltage (I–V) curves were obtained by giving series of 800–1200 ms voltage pulses (from −120 to +60 mV, in 20 mV steps) to the oocyte before ligand superfusion and during the *I_Ka_* plateau elicited by 100 μM Ka.

To obtain Ka concentration–*I_Ka_* amplitude curves, GluK1-2a receptors were activated with different concentrations of Ka. The following form of the Hill Equation (1) was used to fit dose–response data:(1)$$I/I_{Ka}max={\left[{1+\left({EC}_{50}/\left[Ka\right]\right)}^{n_{H}}\right]}^{-1}$$
where *I* is the *I_Ka_* amplitude elicited at a given concentration of Ka ([*Ka*]); *EC*_50_ is the agonist concentration required to halve the maximum *I_Ka_*; *I_Ka_max* is the maximum *I_Ka_* recorded; and *n_H_* is the Hill coefficient.

Steady-state currents attained in NR (measured at the last 100 ms of the pulse) and recorded at each voltage were subtracted from the corresponding currents obtained in presence of 100 µM Ka, in order to compute net I–V curves for *I_Ka_*. These values were then normalized to the *I_Ka_*s recorded at −60 mV for each cell.

### 4.8. Statistical Analysis

Statistical analysis was performed to compare the number of immunoreactive particles between oocyte samples using GraphPad Prism 7 (GraphPad Software, Boston, MA, USA). A *p* < 0.05 was considered statistically significant. All quantitative data were expressed as mean value ± SEM. Asterisks indicate significant differences between experimental groups (* = *p*-value < 0.05; ** = *p*-value < 0.01; ***= *p*-value < 0.001).

For functional experiments, mean ± SEM are presented; the number of oocytes and oocyte donor (frogs) from which the data were obtained is indicated by “n” and “N”, respectively. A Student’s *t*-test was used when contrasting two group means of normally distributed values; otherwise, the Mann–Whitney rank-sum test was applied. When comparing two population proportions, an *χ*^2^-test was used. A significance level of *p* < 0.05 was adopted in all cases.

### 4.9. Drugs

Reagents of general use were purchased from Scharlau Chemie S.A. (Barcelona, Spain); HEPES was purchased from Acros Organics (Geel, Belgium); and MS-222, penicillin, streptomycin, and Ka were purchased from Sigma-Aldrich (St. Louis, MO, USA).

## Figures and Tables

**Figure 1 ijms-24-16852-f001:**
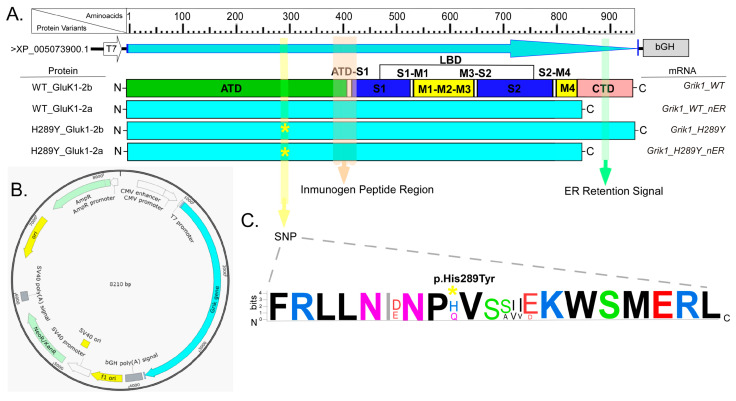
(**A**) Schematic representation of the protein sequences of GluK1, WT and mutated (H289Y). Protein domains are represented with colored bars in GluK1_WT: ATD (amino-terminal domain), S1 and S2 are two segments of the polypeptide chain that together form the ligand-binding domain (LBD), M1–M4 represent transmembrane domains, and CTD is the C-terminal domain. Color arrows show protein regions corresponding to the SNP (H289Y) in the ATD, immunogen peptide region (380–430 aa), and the ER retention sequence (amino acids 841–905). The asterisk (*) in yellow indicates the location of single-point mutation in the protein sequences. (**B**) Schematic representation of the plasmid for in vitro transcription of the distinct variants of *Grik1* gene. (**C**) WebLogo representation of the consensus alignment of the SNP containing region using 50 GluK1 ortholog proteins from different species (78% of conservation), including hamster and humans. It is important to note that the residue at position 289 in WT GluK1 receptor is a histidine (H) in *Mesocricetus auratus*, *Mus musculus*, *Rattus norvegicus*, and *Homo sapiens*. Color code: polar amino acids (G, S, T, Y, C, Q, N) shown in green, basic (K, R, H) in blue, acidic (D, E) in red, and hydrophobic amino acids (A, V, L, I, P, W, F, M) in black. Bits, in the y-axis, indicate the frequency of the corresponding amino acid with the overall height of each stack proportional to the sequence conservation.

**Figure 2 ijms-24-16852-f002:**
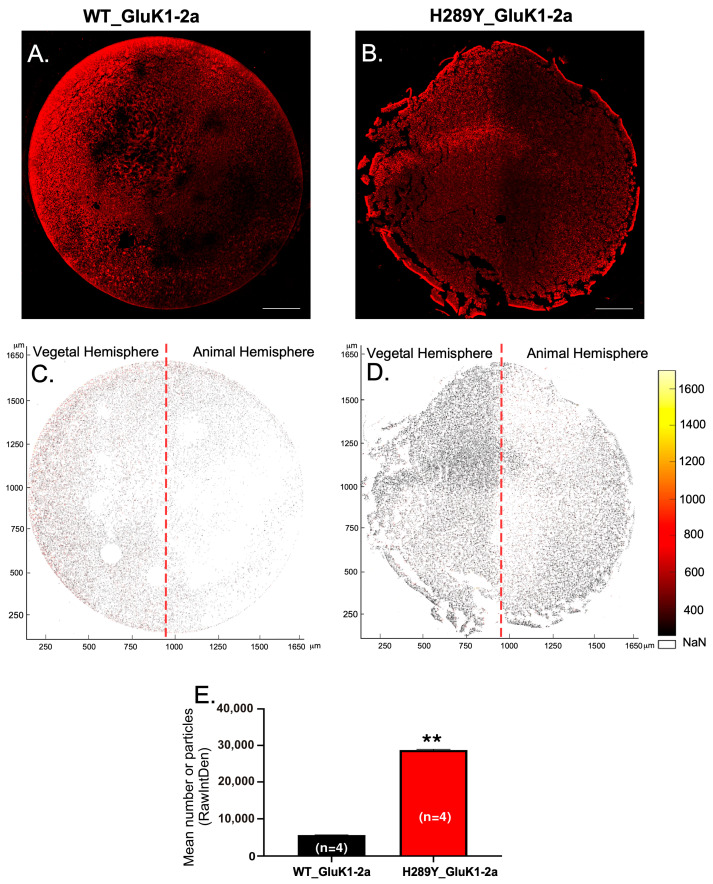
Distribution of GluK1-2a-immunolabeling in *Xenopus* oocytes containing WT GluK1-2a and H289Y GluK1-2a receptors. Representative confocal microscopy images showing GluK1-2a-immunolabeling in cross-sections of oocytes expressing WT GluK1-2a (**A**) and H289Y GluK1-2a (**B**) receptors. Scale bar = 200 μm. MATLAB maps of the oocyte cross-sections corresponding to panels A and B showing comparison of GluK1-2a-immunolabeling between WT GluK1-2a (**C**) and H289Y GluK1-2a (**D**) receptors. Inset in the right of the maps show the optical density-to-color calibration bars. NaN stands for not-a-number values. (**E**) Comparative levels of the sum of the values of the pixels (RawIntDen) corresponding to images (**C**,**D**). Each bar represents the mean number of particles ± hemistandard deviation (SEM). Statistical significance: ** *p* < 0.01.

**Figure 3 ijms-24-16852-f003:**
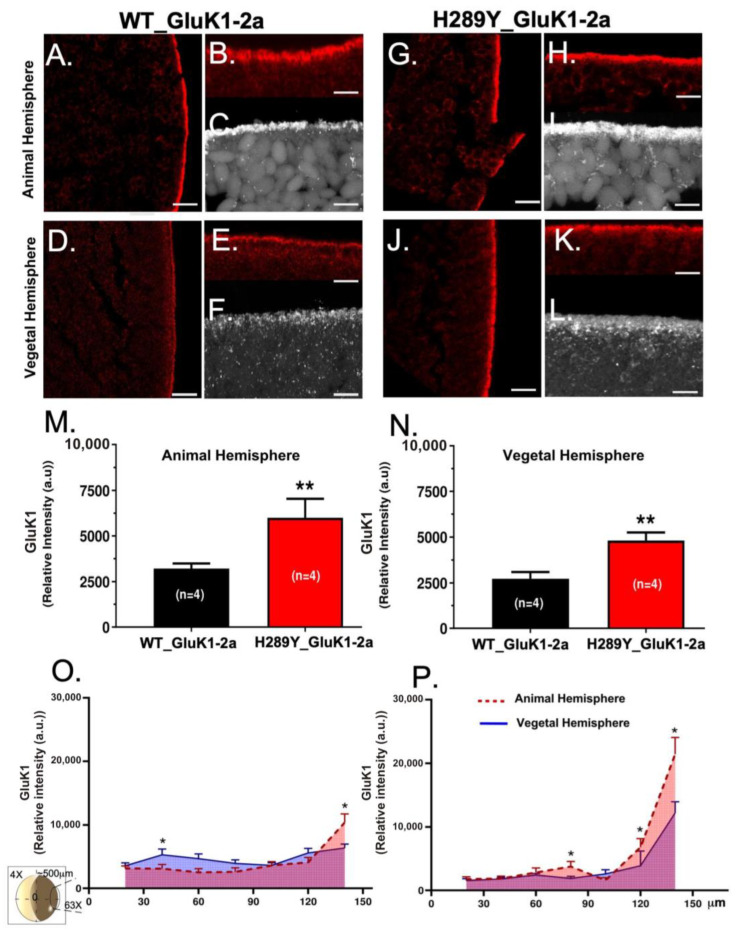
Confocal immunofluorescence micrographs showing representative cross-sections of animal and vegetal oocyte hemispheres. Immunolabeling for WT GluK1-2a (**A**–**F**) and H289Y GluK1-2a (**G**–**L**) receptors. Scale bar = 50 μm in (**A**,**D**,**G**,**J**) and 10 μm in (**B**,**C**,**E**,**F**,**H**,**I**,**K**,**L**). (**M**,**N**) Relative emission intensity levels (a.u.—arbitrary units—) of immunofluorescence for WT GluK1-2a and H289Y GluK1-2a receptors in the animal (**M**) and vegetal (**N**) oocyte hemispheres. Each bar represents the relative intensity ± SEM. (**O**,**P**) Expression levels vs. distribution of WT GluK1-2a (**O**) and H289Y GluK1-2a (**P**) receptors across 150 μm of the oocyte membrane (as indicated in the oocyte drawing in the left part of the panel). Each bar represents the relative intensity ± SEM. Statistical significance: * *p* < 0.05, ** *p* < 0.01.

**Figure 4 ijms-24-16852-f004:**
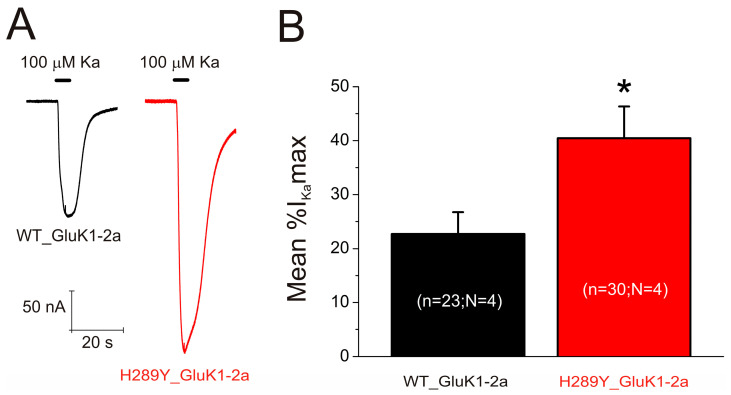
(**A**) Representative *I_Ka_*s obtained through application of 100 µM kainate (Ka) for 5 s in two oocytes from the same donor, one injected with WT GluK1-2a-coding mRNA (left, black recording) and the other with H289Y GluK1-2a-coding mRNA (right, red recording). Henceforth, bars above the traces show the timing of agonist application, downward deflections represent inward currents, and, unless otherwise stated, the holding potential was −60 mV. (**B**) Bar diagram showing the average percentage of the normalized *I_Ka_*s for cells incorporating either WT or H289Y GluK1-2a receptors. The asterisk (*) indicates significant differences between the two values (*p* < 0.05 in Mann–Whitney rank-sum test). “n” and “N” specify the number of oocytes and donors (frogs), respectively.

**Figure 5 ijms-24-16852-f005:**
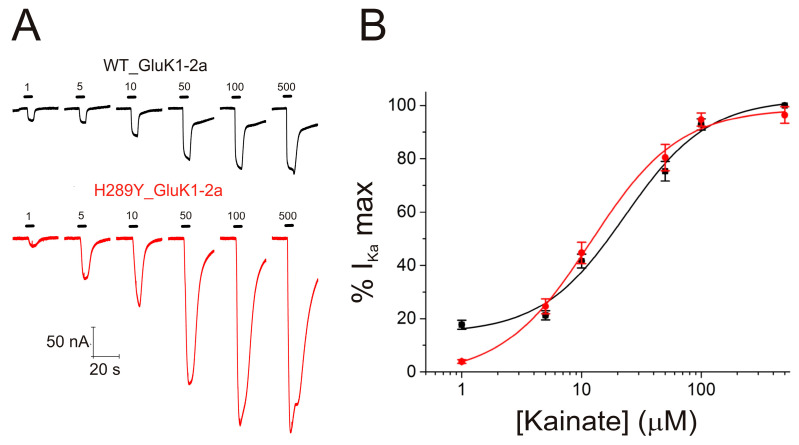
(**A**) Recordings obtained by applying 1, 5, 10, 50, 100, and 500 μM Ka to an oocyte bearing either WT (black recordings) or H289Y (red recordings) GluK1-2aRs. (**B**) Averaged Ka concentration–*I_Ka_* amplitude curves for WT (black solid symbols; n = 4, N = 3) and H289Y (red solid symbols; n = 7, N = 3) GluK1-2aRs. Data were normalized to the maximal *I_Ka_* elicited and the Hill equation was fitted to them (continuous lines).

**Figure 6 ijms-24-16852-f006:**
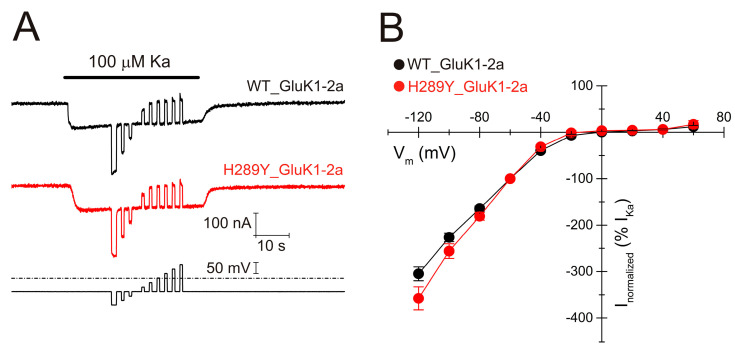
(**A**) *I_Ka_*s evoked using 100 µM Ka when applying voltage pulses from −120 to +60 mV, as shown underneath, in oocytes bearing WT (black recording) or H289Y (red recording) GluK1-2aRs. (**B**) Net I–V relationship of *I_Ka_*s elicited through the protocol shown in A. Black and red symbols are for *I_Ka_*s elicited in oocytes that expressed WT (n = 7; N = 3) or H289Y (n = 6; N = 3) GluK1-2aRs, respectively. Net *I_Ka_*s were normalized as the percentage of the *I_Ka_* obtained at −60 mV.

## Data Availability

Immunofluorescence and electrophysiological data are available on request.

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
