# Peer review of "Enhanced Membrane Incorporation of H289Y Mutant GluK1 Receptors from the Audiogenic Seizure-Prone GASH/Sal Model: Functional and Morphological Impacts on Xenopus Oocytes"

_ijms, 2023, doi:10.3390/ijms242316852_

Round 1

Reviewer 1 Report

Comments and Suggestions for Authors

Reviewer comments and suggestions

The authors in this study discuss epilepsy with the help of an animal model GASH/Sal (genetically audiogenic seizure-prone hamster from Salamanca) exhibits seizures resembling human generalized tonic-clonic convulsions. The authors utilized confocal microscopy in Xenopus oocytes to investigate how the H289Y mutation, compared to the wild-type (WT), affects the expression and cell surface trafficking of GluK1 receptors. Furthermore, they examined the functional effects of the H289Y mutation. The study results indicated that this mutation increases the expression and incorporation of GluK1 receptors into oocyte’s membrane, enhancing kainate-evoked currents without affecting their functional properties.  

Overall, the manuscript needs to be modified. I have listed the concerns and comments that needed to be explained or modified.

  1. Line 43 GluK1-5 can also written in Grik1-5
  2. Line 66-68 Please explain more on this topic
  3. Line 93-94 No need to mention the result part in the introduction section
  4. Lines 130–132 should be clearly understable. please rewrite
  5. Line 161-163 Is there any possible reason for this, please mention
  6. Line 167-169 What does the study indicate
  7. Line 199-200 Is there any reason for this behavior
  8. Line 259-261 In my view these lines need not to be present her, should be deleted.
  9. Line 281-282 The authors did not discuss more on CTD,. please explain wherever required
  10. Line 303-312 Did the authors study the distribution? Please mention the figures
  11. Line 340-342 It seems that this paragraph should be in the introduction section
  12. The journal style should be based on MDPI guidelines.

Author Response

The authors in this study discuss epilepsy with the help of an animal model GASH/Sal (genetically audiogenic seizure-prone hamster from Salamanca) exhibits seizures resembling human generalized tonic-clonic convulsions. The authors utilized confocal microscopy in Xenopus oocytes to investigate how the H289Y mutation, compared to the wild-type (WT), affects the expression and cell surface trafficking of GluK1 receptors. Furthermore, they examined the functional effects of the H289Y mutation. The study results indicated that this mutation increases the expression and incorporation of GluK1 receptors into oocyte’s membrane, enhancing kainate-evoked currents without affecting their functional properties.

            Overall, the manuscript needs to be modified. I have listed the concerns and comments that needed to be explained or modified.

            - Line 43 GluK1-5 can also written in Grik1-5:

            Reviewer is right, we have written Grik1-5.

            - Line 66-68 Please explain more on this topic:

            We have rewritten lines 66-68 slightly expanding the information: “Remarkably, since these receptors have a pivotal role in excitatory neurotransmission, they constitute a key therapeutic target, as their dysfunction causes relevant disorders. Indeed, several alterations in Grik genes have been related to Alzheimer’s disease, Huntington’s chorea, amyotrophic lateral sclerosis, and epilepsy [reviewed in 1, 19]”. Furthermore, we consider that we have sufficiently explained in the Introduction the relationship of GluK1 with epilepsy, which is the main subject of the study. Thus, at the final lines of the second paragraph of the Introduction, it is stated: “The important role played by GluK1Rs at excitatory synapses in the central nervous system might suggest their involvement in epileptogenesis [29]. Indeed, Grik1 has been identified previously as an epilepsy-associated gene [30, 31], and it has been described the relation of Grik1 variants with juvenile absence epilepsy [32] and with sudden unexpected death in epilepsy [33].”

            - Line 93-94 No need to mention the result part in the introduction section:

            We agree with reviewer. Done

            - Lines 130–132 should be clearly understable. Please rewrite

            We have modified and simplified the first lines of the paragraph, now it reads as follows: “Then, we assessed the localization and distribution of WT and H289Y GluK1-2aRs throughout sections of whole oocytes, including the analysis of the distribution of receptors in the animal or vegetal hemispheres of the cells. In these sections, lower immunoreactivity of WT GluK1-2aRs, compared to H289Y GluK1-2aRs, was detected (Fig. 2A, B). Furthermore, analysis of the immunoreactivity of GluK1-2aRs using RawIntDen in MATLAB matches the architecture of Xenopus oocytes.”

            - Line 161-163 Is there any possible reason for this, please mention

            Based on our data, any hypothesis that attempts to explain, at molecular level, the increased incorporation of H289Y GluK1 receptors into the oocyte’s membrane would be pure speculation. As indicated in the Discussion of the new version of manuscript: “further research is needed to understand in detail the molecular mechanisms by which the H289Y mutation increases the expression and incorporation of GluK1-2a receptors into the plasma membrane. This should include studies to analyze the functional relevance of this mutation in a native system to know whether the increased protein expression will contribute to the diseased phenotype or not; and others to decipher the possible effects of the H289Y mutation in the GluK1-2a receptor on its interactions with certain proteins, such as Neuropilin Tolloid-like 1 and Neuropilin Tolloid-like 2 (NETO1 and NETO2, respectively), among others; since GluK1 and GluK2 ATDs have been described to have a differential dependence on NETO proteins, which regulate extrasynaptic and synaptic trafficking [2, 10]”. So, we can just say that the increase in H289Y GluK1-2aR immunoreactivity in oocyte membrane is because this mutation, located in the ATD subunit, could be responsible for targeting of receptors in the membrane. Furthermore, this data is supported by the electrophysiological results and, as indicated in Discussion: “it is known that alterations in the ATD would contribute to changes in regulation of synaptic and surface expression of KARs [7].” And below, at the end of Discussion section, we also stated: “Our hypothesis centres on the notion that the H289Y mutation in the GluK1-2aRs may lead to an increased trafficking of these receptors to neuronal membrane sites within the circuit responsible for seizures, thereby favouring an excitatory imbalance of neuronal activity in critical brain regions within the GASH/Sal model”.

            - Line 167-169 What does the study indicate

            Our study indicates what has been presented in the Results section: “…the expression of GluK1-2aRs was higher in some points of the animal hemisphere as it is shown when juxtaposed with the vegetal hemisphere”. And as we have discussed in the new version of manuscript: “The differential distribution of receptors between oocyte’s hemispheres, like that now reported for WT and H289Y GluK1-2aRs, could be due to an uneven distribution of the translation machinery and subsequent translocation of the proteins to the plasma membrane, or to a pre-existence of some components which facilitate the binding of receptors to the plasma membrane [44].”

            - Line 199-200 Is there any reason for this behaviour

            Xenopus oocytes are a convenient cell model for the detailed study of the properties of foreign functional proteins. They can be incorporated into the oocyte membrane after heterologous expression by microinjection of either their corresponding coding cDNA or mRNA, or proteins purified from native tissue (Limon, A.; Mattei, C. The Xenopus Oocyte: A Tool for Membrane Biology. Membranes 2023, 13, 831. https://doi.org/10.3390/ membranes13100831; Ivorra, I.; Alberola-Die, A.; Cobo, R.; González-Ros, J.M.; Morales, A. Xenopus Oocytes as a Powerful Cellular Model to Study Foreign Fully-Processed Membrane Proteins. Membranes 2022, 12, 986). However, this cell model has few disadvantages; one of them is the difference in effectiveness of the expression or incorporation of the studied proteins when comparing the responses observed from different frog donors. Several reasons have been proposed in order to explain this variability including seasonal variations affecting the frog physiology (Weber, W.M. (1999). Ion currents of Xenopus laevis oocytes: state of the art. Biochim Biophys Acta, 1421, 213-233. Doi: 10.1016/s0005-2736(99)00135-2.; Terhag, J.; Cavara, N.A.; Hollmann, M. (2010). Cave Canalem: how endogenous ion channels may interfere with heterologous expression in Xenopus oocytes. Methods, 51, 66-74. Doi: 10.1016/j.ymeth.2010.01.034) but until now, the real explanation is still missing. Another point to have in mind is that the density and number of functional membrane proteins fluctuate along the time (Morales, A.; Aleu, J.; Ivorra, I.; Ferragut, J.A.; González-Ros, J.M.; Miledi, R. (1995). Incorporation of reconstituted acetylcholine receptors from Torpedo into the Xenopus oocyte membrane. Proc Natl Acad Sci USA, 92, 8468-8472. Doi: 10.1073/pnas.92.18.8468; Stühmer, W.; Parekh, A.B. Electrophysiological recordings from Xenopus oocytes. (1995). In: Single-channel recording; Sakmann, B., Neher, E. Eds.; Springer: Boston, USA; pp. 341-356. Doi: 10.1007/978-1-4419-1229-9_15), so different times after the injection of the exogenous material or different metabolic states of the donor can be followed by differences in effectiveness of this heterologous expression, anyway without altering the functional properties of the protein expressed.

            - Line 259-261 In my view these lines need not to be present her, should be deleted:

            With deep regret, we are not agreeing with reviewer’s comment. These lines contextualise the discussion and focus on the origin of the main topic of study: the H289Y mutation in GluK1 receptors from GASH/Sal epilepsy model.

            - Line 281-282 The authors did not discuss more on CTD, please explain wherever required

            Since the H289Y mutation in GluK1 is located in ATD, we consider more important to discuss about this domain and not so much about CTD. Nevertheless, studies about the importance of CTD in the trafficking of KARs are briefly outlined previously, in the Introduction section: “Alternative splicing of GluK1 CTD can produce four different isoforms named: GluK1-a, GluK1-b, GluK1-c [12], and GluK1-d (which is only found in human [13, 14]). GluK1-b is a larger isoform than GluK1-a, comprising an endoplasmic reticulum (ER) retention motif, whereas GluK1-c includes an additional sequence of 29 residues into the CTD of GluK1-b [15].” So, we have now included in the Discussion: “…intracellular CTD for KARs trafficking and their retention in ER [36-39], as stated above (see Introduction), it is known…”.

            - Line 303-312 Did the authors study the distribution? Please mention the figures

            If the reviewer refers to whether we have studied whether the heterologous proteins (WT vs H289Y GluK1-2a receptors) along the membrane were uniformly distributed or, on the contrary, were found in patches (as commented in Discussion) the answer is no. In the cross-sections of oocytes, at membrane level, we just observed which is stated in the following modified sentence of Results: “…immunoreactivity of H289Y GluK1-2aR seemed to exhibit a distinct pattern, showing a continuous overlay on the membrane, in contrast to WT GluK1-2Ra, which displayed well-defined spherical accumulations of approximately 0.3 µm (Fig. 3C, F, I, L)”. In other hand, we just analysed differences in the number of receptors distributed between animal and vegetal hemispheres. As stated in Discussion now: “In relation with the distribution of GluK1-2aRs in oocyte’s membrane, we have observed that there is a higher presence of receptors in some points of the animal hemisphere plasma membrane”, and this data is related with figure 3O and 3P (values at 140 μm for WT and mutant, and values at 80, 120 and 140 μm). We have now stated in Results: “Remarkably, in both experimental groups (WT and mutated receptors), the expression of GluK1-2aRs was higher in some points of the animal hemisphere when juxtaposed with the vegetal hemisphere (Fig. 3O, P; values at 140 μm for WT and H289Y, and values at 80, 120 and 140 μm for H289Y).”

            - Line 340-342 It seems that this paragraph should be in the introduction section

            Reviewer is right. We have moved these lines to the second paragraph of the introduction.

            - The journal style should be based on MDPI guidelines.

            We have followed the journal style on MDPI guidelines.

Reviewer 2 Report

Comments and Suggestions for Authors

The manuscript by Sandra M. Díaz-Rodríguez et al., titled “Enhanced Membrane Incorporation of H289Y Mutant GluK1 Receptors from the Audiogenic Seizure-Prone GASH/Sal Model: Functional and Morphological Impacts on Xenopus Oocytes," is an interesting study and can be considered for publication with some major edits.

Major Comments

  1. Please provide the extent of conservation of these amino acids in several vertebrate species, including humans.
  2. Using in silico prediction tools (SIFT, PROVEN, iStable, MuStab, etc.), predict whether the impact of this mutation is disease-causing or neutral.
  3. The immunolabelling data should be confirmed using Westen blot analysis for protein surface expression.
  4. Functional relevance must be analysed by expressing this mutated protein in a native system using electrophysiology to know whether the increased protein expression will contribute to the diseased phenotype or not.

Minor comment:

  1. Please explain in discussion the nature of the proteins formed (ex., homotetramers or heterotetramers).
  2. In the method section, please include information regarding the site-directed mutagenesis performed to get the H298Y variant of GluK1.
  3. Please check the typos in Line 107 (‘Based on this data’).

Author Response

The manuscript by Sandra M. Díaz-Rodríguez et al., titled “Enhanced Membrane Incorporation of H289Y Mutant GluK1 Receptors from the Audiogenic Seizure-Prone GASH/Sal Model: Functional and Morphological Impacts on Xenopus Oocytes," is an interesting study and can be considered for publication with some major edits.

            Major Comments

            - Please provide the extent of conservation of these amino acids in several vertebrate species, including humans.

            We appreciate the reviewer’s comment. As he/she knows, in Fig. 1 is showed the consensus alignment of the SNP containing region from 50 GluK1 ortholog proteins from different species. It is important to note that these species include Homo sapiens and this region has a percentage of conservation equal to 78%. Furthermore, as shown in the attached figure below, the residue at position 289 in WT GluK1 receptor is a histidine (H) in Mesocricetus auratus, Mus musculus, Rattus norvegicus and Homo sapiens. We have now included in the legend of Fig. 1 the following: “…from different species (78% of conservation), including hamster and humans. It is important to note that the residue at position 289 in WT GluK1 receptor is a histidine (H) in Mesocricetus auratus, Mus musculus, Rattus norvegicus and Homo sapiens”.

            - Using in silico prediction tools (SIFT, PROVEN, iStable, MuStab, etc.), predict whether the impact of this mutation is disease-causing or neutral.

            We appreciate the insightful suggestion of the Reviewer regarding the utilization of in-silico prediction tools to assess the potential disease-causing or neutral effects of the missense single-nucleotide polymorphism (C9586732T, p.His289Tyr) in the Grik1 gene. We totally agree with the importance of employing such tools to gain a comprehensive understanding of the mutation's impact.

             In response to the reviewer's recommendation, we want to inform that we conducted additional in-silico experiments in a separate study titled "Delving into the Significance of the His289Tyr Single-Nucleotide Polymorphism in the Glutamate Ionotropic Receptor Kainate-1 (Grik1) Gene of a Genetically Audiogenic Seizure Model", which is currently under revision at Frontiers in Molecular Neuroscience. In this research, we extensively explored the effects of this sequence variation within the amino-terminal domain using advanced in-silico 3D modelling techniques and computational stability predictors. Our methodology involved a two-step process, wherein we employed the highly regarded protein structure prediction tool AlphaFold2 to predict the 3D structure of the GluK1 receptor. Furthermore, we used PyMOL to calculate interatomic interactions in both wild-type and mutated GluK1 receptor residues, providing valuable insights into the structural environment of the mutation. To assess the impact of the p.H289Y variant on the protein's thermodynamic stability (ΔΔGStability), we employed several computational stability predictors, namely Dynamut2, INPS3D, FoldX, and MAESTRO. Each predictor considered different aspects of the mutation's effect on protein stability, including dynamics, physicochemical properties, and empirical force fields. Notably, all stability predictors consistently indicated that the p.H289Y variant led to protein stabilization, as discussed in detail in the submitted paper.

             Following the reviewer’s suggestion, we have now incorporated this essential information into the discussion section of the manuscript, specifically in the paragraph which reads: "In addition to the study presented here, our research group conducted in-silico experiments utilizing protein 3D structure modelling and various computational stability predictors. These experiments indicated that the single-point mutation H289Y likely leads to protein stabilization by increasing intermolecular interactions, as compared to the wild-type GluK1 receptor [50]. Concurrently, we observed a concentration of GluK1-immunoreactivity near the cell nucleus in brain structures associated with the GASH/Sal seizure neuronal network [50]. These combined findings support the assertion made in our study that the mutation has the potential to alter the GluK1 trafficking mechanism while exerting a relatively minor influence on the functional properties of the receptor. The precise genetic alterations and molecular mechanisms contributing to audiogenic susceptibility in the GASH/Sal strain remain an ongoing area of investigation, necessitating further research. Our hypothesis centres on the notion that the H289Y mutation in the GluK1-2aRs may lead to an increased trafficking of GluK1-2aRs to neuronal membrane sites within the circuit responsible for seizures, thereby favouring an excitatory imbalance of neuronal activity in critical brain regions within the GASH/Sal model.” (See discussion section in the revised version of the manuscript).

            We hope that these additions adequately address your concern. Additionally, please find below the reference, to our separate study that provides a comprehensive examination of the mutation's effects:

Díaz-Rodríguez, S.M., Herrero-Turrión, M.J., García-Peral, C., and Gómez-Nieto, R. "Delving into the Significance of the His289Tyr Single-Nucleotide Polymorphism in the Glutamate Ionotropic Receptor Kainate-1 (Grik1) Gene of a Genetically Audiogenic Seizure Model." Under revision in Frontiers in Molecular Neuroscience. (We have included this study in the reference list, as[50], of the revised version of the manuscript following favourable feedback from the reviewers, as we hope it will be published shortly).

            The following sentence has been modified accordingly:

            Previous version lines 397-395. “This antibody has been effectively employed in our ongoing research on the GASH/Sal brain (unpublished data)”.

            Revised version lines: “This antibody has been effectively employed in our ongoing research on the GASH/Sal brain [50]”.

            - The immunolabelling data should be confirmed using Westen blot analysis for protein surface expression.

            We appreciate the reviewer's suggestion regarding the use of Western blot analysis to confirm the immunolabeling data for protein surface expression. While we acknowledge the value of Western blot analysis, we respectfully hold a different viewpoint. In our assessment, we believe that Western blot analysis is not essential to substantiate the primary discovery of our study, which focused on the impact of the H289Y mutation in GluK1-2a receptors from GASH/Sal hamsters. Our study unequivocally demonstrates an increase in the expression and incorporation of GluK1 receptors into the oocyte's membrane. This finding, supported by our immunolabeling data, remains robust and sufficient for our study's objectives. Our disagreement with this comment can be attributed to three main reasons:

            Firstly, Western blotting provides information about the presence and amount of protein in the whole oocyte or in its membrane, but not on the existence of functional receptors incorporated into the membrane. In our opinion, the latter is of paramount importance in determining whether the mutation present in the GASH/Sal hamster has a genuine implication in epileptogenesis. For this reason, the electrophysiological recordings of our study are unquestionable evidence (in agreement with the quantitative immunolabelling data) that confirm the increase of H289Y GluK1-2a receptors (vs. WT receptors) in the oocyte membrane, based on the rise in the functional expression and in the percentage of maximum IKa obtained by two-electrode voltage-clamp technique in oocytes bearing H289Y GluK1-2aRs.

             Secondly, the immunolabelling data of our study was analysed quantitatively by designing a script in the MATLAB software (© MATLAB R-2017, Scatterplot function) that allowed us to quantify RawInDet. As stated in Materials and methods section: “The image analysis was conducted using ImageJ software (version 1.51g-v1.51n; Fiji package), an open-source tool routinely employed by our research group [57, 58]. In this current image analysis, a comprehensive assessment of various parameters was undertaken. This encompassed the quantification of the product of area and mean gray value (IntDen), the summation of pixel values (RawIntDen), determination of particle area, and computation of mean and mode values in both X and Y coordinates”. So, as mentioned, the RawInDet dataset was visually presented to illustrate quantitative distinctions between WT GluK1-2aRs and H289Y GluK1-2aRs in sections of whole oocytes (Fig. 2E). Moreover, we measured in confocal immunofluorescence micrographs, the relative emission intensity levels of immunofluorescence for WT GluK1-2a and H289Y GluK1-2a receptors in the animal and vegetal hemisphere membranes (Fig. 3M, N). As stated in the manuscript, there was an increase of immunoreactivity for H289Y GluK1-2aRs (vs WT GluK1-2aRs). We have included now, in the new version of the manuscript, the numerical values presented in Figs. 2E, 3M and 3N.

       Thirdly, while immunoblotting typically functions as a valuable tool for confirming protein levels, in these specific experiments, it requires the analysis of a substantial number of oocytes to attain statistically significant results. Unfortunately, given the inherent limitations of our experimental system, obtaining a sufficient quantity of oocytes to meet the statistical significance criteria has become an exceedingly challenging and unfeasible task at this moment.

            Finally, we consider important to highlight that in the complementary study mentioned above, which will be published soon (Díaz-Rodríguez et al., "Delving into the Significance of the His289Tyr Single-Nucleotide Polymorphism in the Glutamate Ionotropic Receptor Kainate-1 (Grik1) Gene of a Genetically Audiogenic Seizure Model." Under revision in Frontiers in Molecular Neuroscience), we have employed a multi-technique approach, including Western blotting but also gene expression analysis (RT-qPCR) and immunohistochemistry with bright-field and confocal microscopy, to investigate critical seizure-associated brain regions in GASH/Sal animals under seizure-free conditions compared to matched wild-type controls. We have detected disruptions in the transcriptional profile of the Grik1 gene within the audiogenic seizure-associated neuronal network. We have also observed alterations in GluK1 protein levels in various brain structures, accompanied by an unexpected lower molecular weight band in the inferior and superior colliculi. This has been correlated with substantial disparities in GluK1-immunolabeling distribution across multiple brain regions, including the cerebellum, hippocampus, subdivisions of the inferior and superior colliculi, and the prefrontal cortex.

            - Functional relevance must be analysed by expressing this mutated protein in a native system using electrophysiology to know whether the increased protein expression will contribute to the diseased phenotype or not.

            We appreciate the reviewer's comment and find the proposed experiment very interesting, but in our humble opinion this is not the scope of the present study. As indicated, our research has been focused on the study of morphological and functional effects of H289Y mutation in GluK1 receptors by using Xenopus oocytes as cellular model. As stated recently (Limon, A.; Mattei, C. The Xenopus Oocyte: A Tool for Membrane Biology. Membranes 2023, 13, 831.Membranes, 2023) the Xenopus oocyte is a very practical system in the toolbox of scientists who wish to have a cellular model for expressing genes of interest. Moreover, Xenopus oocytes have also become the reference model for biophysical studies dealing with structure to function relationships, addressed to unravel the functional consequences of specific mutations of membrane receptors, channels, and transporters (Ivorra, I.; Alberola-Die, A.; Cobo, R.; González-Ros, J.M.; Morales, A. Xenopus Oocytes as a Powerful Cellular Model to Study Foreign Fully-Processed Membrane Proteins. Membranes 2022, 12, 986). In this context, the present study can therefore serve as a starting point for further research on native tissues/cells. So, we have included, at the end of the second paragraph of the Discussion section, the following: “This should include studies to analyze the functional relevance of this mutation in a native system to know whether the increased protein expression will contribute to the diseased phenotype or not”.

            Minor comments:

            - Please explain in discussion the nature of the proteins formed (ex., homotetramers or heterotetramers).

            Reviewer is right. We have included in the first paragraph of Discussion: “These receptors, after the microinjection of their coding mRNA, were expressed by the oocytes and incorporated in their membrane as homotetramers

            - In the method section, please include information regarding the site-directed mutagenesis performed to get the H298Y variant of GluK1.

            As described in the methods section, cDNA encompassing either the WT version or the H298Y variant were obtained by the gene synthesis service provided by GenScript. To further clarify this issue now the methods reads as follow: “Grik1 cDNA fragments, bearing either the wild-type sequence or the missense point mutation H289Y, both synthesized and cloned into the pcDNA3.1 vector (Invitrogen, USA) by GenScript gene synthesis service, yielded, respectively, plasmids….”

            - Please check the typos in Line 107 (‘Based on this data’).

            We have been replaced “based in this data” by “based on this data”.

            Thank you for your valuable input, which has undoubtedly enhanced the quality and comprehensiveness of our research.

Round 2

Reviewer 2 Report

Comments and Suggestions for Authors

The previous concerns still stay the same.

The authors need to address them

1. Using in silico prediction tools (SIFT, PROVEN, iStable, MuStab, etc.),
predict whether the impact of this mutation is disease-causing or neutral.

2. The immunolabelling data should be confirmed by using Westen blot analysis for protein surface expression.

Author Response

            First of all, thanks for the suggestions and for the time spent. We sincerely appreciate your comments, which we believe have served to improve the manuscript.

Question 1-   

The previous concerns still stay the same.

            The authors need to address them:

            - Using in silico prediction tools (SIFT, PROVEN, iStable, MuStab, etc.), predict whether the impact of this mutation is disease-causing or neutral.

Answer:         As we stated in our previous answer, we totally agree with the importance of employing such tools to gain a comprehensive understanding of the mutation's impact on the protein. Nevertheless, also is important to note that this kind of tools is merely a PREDICTION (as reviewer mention) and constitute a complementary instrument that cannot substitute experimental data obtained in the laboratory, as our electrophysiological results. Our data unequivocally confirm that H289Y GluK1 receptors are functional and, therefore, that these proteins are stable. And this is of paramount importance. Besides, the silico tools SIFT and iStable, do not predict whether the impact of this mutation is disease-causing or neutral, as stated by the reviewer. The software platforms indicate for SIFT and iStable the following: “(SIFT) predicts whether an amino acid substitution affects protein function based on sequence homology and the physical properties of amino acids”, and “iStable” is an integrated predictor for protein stability change upon single mutation”. It is right that the alteration of function and/or stability of the protein by a single-point mutation could be responsible (or the exclusive cause) of a specific disease but, in the case of epilepsy, this is a very remote possibility, since it is a multifactorial disease (Ferraro TN, Dlugos DJ, Buono RJ. Role of genetics in the diagnosis and treatment of epilepsy. Expert Rev Neurother. 2006; 6(12):1789-800; Witt JA, Becker AJ, Helmstaedter C. The multifactorial etiology of cognitive deficits in epilepsy and the neuropathology of mesial temporal lobe epilepsy beyond hyperphosphorylated tau. Alzheimers Dement. 2023; 19(7):3231-3232).

            Moreover, we reiterate (once again) that to assess the impact of the p.H289Y variant on the protein's thermodynamic stability (ΔGStability), we employed several computational stability predictors, namely Dynamut2, INPS3D, FoldX, and MAESTRO, in the separate study entitled "Delving into the Significance of the His289Tyr Single-Nucleotide Polymorphism in the Glutamate Ionotropic Receptor Kainate-1 (Grik1) Gene of a Genetically Audiogenic Seizure Model", which is currently under revision at Frontiers in Molecular Neuroscience. Notably, all stability predictors consistently indicated that the p.H289Y variant led to protein stabilization, as discussed in detail in the submitted paper.

            Despite all of this, and due to the insistence of the reviewer, we have now used the prediction tools SIFT and iStable. Both indicated (in accordance with our electrophysiological results obtained in the present work, and with the tools Dynamut2, INPS3D, FoldX, and MAESTRO, used in the complementary study) that H289Y amino acid substitution is tolerant (SIF Score > 0.05) and does not decrease the stability of the protein (iStable Conf. Score = 0.56).

            We have included in Results and Materials and Methods sections the following paragraphs:

            - Results: “Finally, to assess the impact of the H289Y mutation on the function and thermodynamic stability of GluK1Rs, we employed the two in silico predictors SIFT and iStable. In accordance with our electrophysiological data, the prediction analysis for the H289Y mutation suggests that it is tolerant (SIF Score > 0.05) and does not decrease the stability of the protein (iStable Conf. Score = 0.56).”

            - Materials and Methods:4.7. In silico prediction tools

            In silico tools were applied to identify the potential functional impact on GluK1 of H289Y mutation. In silico was assessed using the online sequence homology-based tools SIFT (https://sift.bii.a-star.edu.sg/index.html) version 6.2.1, and iStable (http://predictor.nchu.edu.tw/iStable/).”

Question 2-   

            - The immunolabelling data should be confirmed using Westen blot analysis for protein surface expression.

Answer:         We respectfully continue to hold a different viewpoint and we believe that Western blot analysis is not essential to substantiate the primary discovery of our study, which focused on the impact of the H289Y mutation in GluK1-2a receptors from GASH/Sal hamsters. As we stated in our previous answer: “our study unequivocally demonstrates an increase in the expression and incorporation of GluK1 receptors into the oocyte's membrane”. And this finding is supported by our immunolabeling and electrophysiological data, which remains robust and sufficient for our study's objectives. We understand that the information discovered by a Western blot analysis would corroborate but not surpass that given by the two techniques we have used in this manuscript. Moreover, 10 days (the period granted by editor) is insufficient to carry out the proposed analysis.

In the revised manuscript, we have highlighted all modifications.

We hope to have answered the questions raised and you agree with the new version of the manuscript.

Best regards

Dolores E. López

Round 3

Reviewer 2 Report

Comments and Suggestions for Authors

I still believe major comments need to be addressed

Author Response

Attending editor comments, we have explained (in the final part of the Discussion section) the limitation of western blot and the reason for not including this technique in the current study. Additionally, we have rearranged the predictive modelling paragraph and located it at the beginning of Results section, highlighting the scientific rationale that in silico analysis are of predictive value. All the revisions are clearly highlighted, for that we have used the "Track Changes" function of Microsoft Word. Finally, due to an inclusion of a new citation related with prediction tools, some references are renumbered.